# Hyperpolarized Xenon-129 MRI: Narrative Review of Clinical Studies, Testing, and Implementation of Advanced Pulmonary In Vivo Imaging and Its Diagnostic Applications

**DOI:** 10.3390/diagnostics15040474

**Published:** 2025-02-16

**Authors:** Jamie L. MacLeod, Humam M. Khan, Ava Franklin, Lukasz Myc, Yun Michael Shim

**Affiliations:** 1Division of Pulmonary and Critical Care, Department of Medicine, University of Virginia, Charlottesville, VA 22908-0546, USA; jlm6re@uvahealth.org (J.L.M.); zth3qp@virginia.edu (A.F.); 2North Alabama Medical Center, 1701 Veterans Dr., Florence, Al 35630, USA; hkhan@namccares.com (H.M.K.); lmyc@namccares.com (L.M.)

**Keywords:** hyperpolarized xenon-129 MRI, hyperpolarized gas MRI, respiratory disease, pulmonary disease, diagnostic test

## Abstract

Hyperpolarized xenon-129 MRI (^129^XeMRI) has emerged as a powerful tool in the identification, evaluation, and assessment of disease endotyping and in response to interventions for a myriad of pulmonary diseases. Growing investigative efforts ranging from basic science to application in translational research have employed ^129^XeMRI in the evaluation of pulmonary conditions such as chronic obstructive pulmonary disease (COPD), idiopathic pulmonary fibrosis (IPF), asthma, and cystic fibrosis (CF). The novel feature of ^129^XeMRI is its ability to generate anatomic and physiologic readouts of the lung with resolution from the whole lung down to the lobar level. Additional advantages include being non-invasive and non-radioactive, and utilizing an inexpensive and ubiquitous noble gas as an inhalation contrast agent: xenon-129. In this review, we outline the clinical advances provided by ^129^XeMRI among common pulmonary diseases with high healthcare burdens in recent decades.

## 1. Overview of ^129^XeMRI

Pulmonary function testing (PFT) and chest computed tomography (CT) have several limitations in detecting early pulmonary physiologic changes with good sensitivity. PFT parameters reflect global lung function and are insensitive to early subtle regional changes. Chest CT visualizes anatomic abnormalities but cannot fully assess early pathologic physiology in the lung. Single-photon emission CT (SPECT) and lung scintigraphy are nuclear medicine imaging techniques that provide lung ventilation and perfusion data but at a much lower resolution than hyperpolarized xenon-129 MRI. ^129^XeMRI was developed to fill these gaps by comprehensively and simultaneously interrogating perturbations occurring in multiple pulmonary microcompartments: airways (“gas”, reflecting ventilation–airflow); interstitial tissues (“tissue or membrane”, reflecting parenchymal tissue integrity related to gas exchange capacity); and red blood cells (RBCs) in pulmonary capillaries (“RBC”, reflecting capillary perfusion) (Figure 1) [1]. With a single breath-hold of hyperpolarized xenon-129, pixel-based ratio maps can be obtained to quantify xenon movement from airways to tissues and finally to RBCs, allowing assessments of global and regional pulmonary airflow and gas exchange physiology. Areas of the lung with vs. without xenon-129 in the airways can be quantified using the ventilation defect percent (VDP), reflecting the severity of obstructive lung diseases [2,3]. The calculated ratios of xenon-129 in pulmonary microcompartments closely reflect biologically important lung physiology: (1) tissue integrity and alveolar surface-to-volume ratio; (2) overall gas exchange efficiency from the airway to the blood; and (3) pulmonary capillary blood flow. Detecting subtle changes in lung function in these pulmonary microcompartments can be highly desirable to determine the natural course, disease endotyping, and therapeutic responses in pulmonary diseases [4].

## 2. COPD

A growing body of research has tested ^129^XeMRI use in evaluating COPD, reporting a robust inter-modality correlation with several conventional diagnostic tools. Previously published work has demonstrated a strong negative correlation between the ventilation defect percentage (VDP) and PFT Forced Expiratory Volume in one second (FEV1) percent predicted, and membrane-to-gas and RBC-to-gas indices, as well as a strong positive correlation with the percent of lung tissues with emphysema quantified by chest computed tomography (CT) and % DLCO, respectively [4]. The clinical relevance of the VDP has been solidified with published confirmation of a strong correlation between the VDP and FEV1 percent predicted, r = −0.80 [4]. Similar correlations have been reported for the ^129^XeMRI-derived apparent diffusion coefficient (ADC), a surrogate marker reflecting alveolar size, which, as reported by Kaushik and colleagues, negatively tracked with FEV1 and Forced Expiratory Volume in one second to Forced Vital Capacity (FEV1/FVC), as well as diffusion capacity of the lung for carbon monoxide divided by alveolar volume (DLCO/VA) [5]. Consistent with these findings, apparent diffusion coefficient (ADC) values were higher in subjects with COPD than in healthy volunteers and age-matched controls, indicating more significant lung parenchymal destruction among the diseased group [5].

While these reports anchor ^129^XeMRI to a familiar framework of thinking about COPD, novel approaches supporting its application in early disease detection and higher-resolution phenotyping have also been studied. In a 2019 study of 19 subjects, Ruppert et al. reported on the ability of XeMR spectroscopy to detect significant changes in septal wall thickness among otherwise healthy current and former smokers when compared to their age-matched non-smoking counterparts, suggesting that this metric may be a sensitive marker of early-stage lung disease in smokers [6]. Other groups reported ^29^XeMRI’s sensitivity to mild emphysema detectable by CT and age-related changes [7]. Similarly, other researchers have noted how aging, smoking, and COPD collectively lead to progressive changes in the lungs, with ^129^XeMRI enabling the visualization of these alterations, thereby offering a more comprehensive evaluation of the disease compared to traditional methods like CT or PFT [8].

In addition to showing promise in early disease detection, ^129^XeMRI’s ability to (1) distinguish COPD from other forms of heterogeneous lung disease and (2) discriminate different phenotypes of COPD from one another has been highlighted in several studies. Guan and colleagues, reporting on a cohort of 45 subjects, published the finding that patients with COPD and cystic fibrosis exhibited significantly more ventilation defects than individuals with idiopathic pulmonary fibrosis (IPF) and healthy individuals, and these defects were highly correlated with the percent predicted FEV1 (R = −0.74) [9]. Patients with COPD and IPF showed elevated tissue/RBC ratios, prolonged RBC T2* relaxation times, and increased RBC chemical shifts. This indicates that three-dimensional single-breath chemical shift imaging ^129^XeMRI can effectively characterize lung diseases, monitor treatment responses, and predict disease outcomes [9]. In work published by our group in 2021, we reported data suggesting that ^129^XeMRI may not only discriminate between emphysematous and non-emphysematous COPD but also provide higher-resolution phenotyping beyond the conventional, binary disease categories of emphysema and chronic bronchitis. Specifically, we described a group of subjects with COPD demonstrating impairment in gas exchange not attributable to emphysema, and ^129^XeMRI uniquely enabled regional analysis within the lung [4]. Building on this idea, Mummy and colleagues reported the ability to map regional VDP changes to changes in RBC transfer in response to bronchodilator therapy, detailing how ventilated regions and RBC transfer defects correlated and highlighted microvascular abnormalities not detected by spirometry [10]. These studies have supported the idea that ^129^XeMRI can provide data comparable to spirometry and chest CT. Its utility and novelty lie in providing multidimensional data on structure and function synchronously and regionally. This offers a significant advance in studying the treatment responses of heterogeneous lung diseases like COPD. Such granular characterization of disease and treatment response holds the promise of personalized and novel therapies for this common lung disease with high healthcare burdens (Table 1).

## 3. Asthma

As another obstructive lung disease, it is not surprising that results in asthma using ^129^XeMRI have been as demonstrative as those in COPD, with a good correlation between ^129^XeMRI and gold-standard assessment tools, such as PFT and spirometry. In a study of 76 participants, Ebner and colleagues reported that ventilation defect scores (VDSs) were notably higher in patients with asthma than in healthy individuals, correlating with the disease severity assessed by PFT [13]. In another study by the same group, young asthmatics exhibited higher ventilation defect percentages (VDPs) than their healthy counterparts, with the VDP increasing with age [14]. Tempering the idea that ^129^XeMRI is a panacea, Mussell and colleagues demonstrated that while the VDP in asthmatics correlated well with FEV1, FEV1/FVC, and Forced Expiratory Flow between 25% and 75% (FEF25–75%), blood eosinophil counts and symptom severity tracked with spirometry but not MRI-derived ventilation metrics [15].

Similarly to in COPD, several studies have suggested the capability of ^129^XeMRI as a more sensitive modality in detecting early disease and previously unrecognized pathophysiology in asthma. A recent study by Qing et al. found differences in gas transfer and ventilation between younger asthmatics and healthy controls, with ^129^XeMRI revealing previously undetected physiological defects at the airspace–interstitial–RBS interface [11]. Additionally, ^129^XeMRI and 3HeMRI may yield somewhat divergent results in detecting ventilation abnormalities in asthmatics, particularly before bronchodilator treatment [16]. Kooner et al. stress that some problems, including ventilation deficiencies, airway inflammation, smooth muscle dysfunction, remodeling, and luminal occlusions in patients with asthma, can be detected reliably by xenon-129 MRI. The ability to view the passage of inhaled gas over the alveolar–capillary barrier to red blood cells makes this imaging approach very useful since it enables researchers to evaluate gas exchange and airflow disturbance directly [17].

The application of ^129^XeMRI in asthma may also extend to furnishing disease-specific diagnostic criteria, monitoring response to treatment, and guiding therapeutic interventions. For example, Peiffer and colleagues reported that in comparison with COPD, ^129^XeMRI revealed deviations in red blood cell-to-tissue/plasma ratios and demonstrated higher spatial resolution than traditional imaging techniques like SPECT in subjects with asthma [18]. In children with asthma, ^129^XeMRI effectively identified ventilation abnormalities and detected post-bronchodilator improvement, aligning with results from spirometry and other pulmonary function tests [19]. In a 2021 study by Lin and colleagues, children with asthma were reported to have more significant percentages of ventilation abnormalities than healthy controls, and those with more defects had greater corticosteroid and healthcare utilization. These findings suggest that ^129^XeMRI may be a helpful diagnostic technique for identifying children more susceptible to asthma exacerbations [20]. Perhaps most illustrative of the technology’s potential impact on therapeutic decision-making was a pilot study of 30 asthmatic subjects randomized to ^129^XeMRI-guided or standard bronchial thermoplasty, in which Chase et al. reported the similar short-term efficacy of the two strategies, with the patient-centered benefit of fewer treatment sessions and peri-procedural events among subjects randomized to the ^129^XeMRI-guided treatment group [21]. In 2023, Svenningsen et al. reported one of the most exciting clinical applications of the ^129^XeMRI ventilation scan, in which complex patterns of therapeutic responses in asthmatic lungs were visualized after a 16-week treatment with a new interleukine-4 receptor alpha-blocking biologic, dupilumab [41]. This study revealed that asthmatics’ therapeutic responses are significantly more complex than previously appreciated, highlighting the advantage of ^129^XeMRI.

## 4. Cystic Fibrosis (CF)

Several investigators have evaluated the effectiveness of ^129^XeMRI in assessing lung function in patients with CF. Not surprisingly, ^129^XeMRI is reliable in measuring the ventilation defect percent (VDP) in the CF population, with consistent results across different institutions and a strong correlation with PFT parameters such as FEV1 and the lung clearance index (LCI) [9,22]. Moreover, Marshall et al. report that ^129^XeMRI was more sensitive than traditional spirometry in detecting early-stage lung disease, with ^129^XeMRI having a robust association with LCI [23]. ^129^XeMRI may likewise be more effective than FEV1 at differentiating patients with CF from healthy controls, particularly in pediatric patients, and has been shown to identify ventilation problems in patients with normal FEV1 values [24,25,26]. Additionally, ^129^XeMRI is effective in determining disease progression and bronchial abnormalities linked to future pulmonary exacerbations, with similar VDPs being reported in a two-center trial, showing promise for multi-center studies, which may subsequently guide clinical decision-making in managing patients with early CF lung disease [25]. These observations have led to an ongoing multi-center clinical trial (BEGIN Novel ImagiNG Biomarkers, ClinicalTrials.gov ID: NCT05517655). In a disease that significantly affects the pediatric population, it is also noteworthy that ^129^XeMRI is non-radioactive and non-invasive while still providing invaluable experimental and clinically meaningful data.

## 5. Idiopathic Pulmonary Fibrosis (IPF)

^129^XeMRI has garnered significant interest in the evaluation of idiopathic pulmonary fibrosis (IPF). Research has indicated robust associations between the current gold-standard PFTs and ^129^XeMR-derived indices. Wang et al. reported a strong correlation between ^129^XeMRI gas transfer measures and DLCO in subjects with IPF [27]. Similarly, Hahn and colleagues demonstrated an association between the red blood cell (RBC)-to-membrane ratio and DLCO [28]. Such results demonstrate that ^129^XeMRI-derived data may reflect in vivo pathophysiologic disturbances in IPF. Aside from correlation with conventional PFT readouts, ^129^XeMRI also furnishes important data to explore and hopefully elucidate IPF etiology. Not only can ^129^XeMRI map lung function in three dimensions and provide a regional assessment of the gas exchange deficit in IPF [29], but ^129^XeMRI spectroscopy can detect interstitial thickening in patients with IPF even when their spirometry was normal, indicative of an ability to identify early disease [12]. Such sensitive detection extends to monitoring and predicting long-term therapeutic responses, as after a year of treatment with antifibrotic drugs, patients with IPF showed improvements in ^129^XeMRI-assessed regional gas exchange [28]. At the same time, conventional PFTs revealed no discernible alterations [28]. Chan et al. observed changes in the ADC and mean acinar dimension before appreciable changes in PFTs, suggesting that early indicators of disease progression in patients with IPF are evaluable by ^129^Xe diffusion-weighted MRI [30]. The capacity of ^129^XeMRI to offer comprehensive regional data may provide opportunities for individualized treatment strategies in IPF. The potential of ^129^XeMRI to detect patterns of IPF disease development was examined by Stiefer et al. [31]. In this study, newly diagnosed patients with IPF were divided into four groups based on MRI assessments of lung tissue uptake and blood transfer (RBC transfer) after undergoing ^129^XeMRI before treatment. Only individuals with high tissue uptake or reduced RBC transfer showed significant decreases in lung function during a 1.5-year follow-up period. This suggests that ^129^XeMRI could be a valuable technique for predicting the course of IPF and individualizing treatment based on the physiologic endotyping enabled by ^129^XeMRI [31].

## 6. COVID-19

In the wake of the COVID-19 pandemic, several studies were designed to investigate the effects of COVID-19 on lung function and persistent symptoms using ^129^XeMRI. Stiefer and colleagues reported that patients with post-acute COVID-19 syndrome (PACS) and long COVID exhibited significant abnormalities in lung function, including impaired gas exchange and ventilation defects, even when structural lung imaging appeared normal [31]. Another study suggested that ^129^XeMRI had sufficient sensitivity to detect persistent lung function impairments and gas transfer defects up to a year after infection [32,33], supporting an objective role for longitudinal disease monitoring. Kooner et al. reported that these lung abnormalities were more closely associated with exercise capacity and performance than self-reported symptoms [34]. Matheson et al. evaluated vascular density and lung function in individuals with post-acute COVID-19 syndrome (PACS) using ^129^XeMRI. This study offered a unique understanding of lung impairment in patients with PACS by assessing the red blood cell (RBC) ratio for gas transfer and CT blood vessel volume in vessels less than 5 mm^2^ (BV5) for small blood vessel density and spirometry and symptom surveys [35]. ^129^XeMRI demonstrated its superiority over conventional imaging techniques in identifying and treating COVID-19-related respiratory problems by revealing severe lung and vascular abnormalities associated with chronic symptoms like dyspnea and decreased exercise capacity [35]. Moreover, ^129^XeMRI has effectively identified various pulmonary phenotypes in patients with long COVID, suggesting diverse pathophysiological profiles [36]. Finally, the early ventilation defect extent assayed by ^129^XeMRI may be tracked with long-term clinical outcomes, indicating a potential role of ^129^XeMRI in the prognostication of COVID-related outcomes [37].

## 7. Miscellaneous Diseases

The use of electronic cigarettes has rapidly increased in the past decade, and ^129^XeMRI can be an excellent modality to define early potential pathologies. One small pilot study has reported a possible dysregulation of pulmonary vascular flow, which requires additional confirmation [42]. Investigators have explored using ^29^XeMRI to detect lung abnormalities in lymphangioleiomyomatosis (LAM) [38] and bronchopulmonary dysplasia (BPD) [39]. Rankine et al. assessed the pathophysiologic impacts of radiation therapy on the lung using ^129^XeMRI and highlighted changes in ventilation and gas transfer [40]. Despite a relative paucity of applied ^129^XeMRI studies in rare lung diseases, acquiring multidimensional, high-resolution, regional (i.e., lobar) data may prove particularly useful in rare lung diseases with varied geographical distribution (i.e., cystic lung disease).

## 8. Conclusions

There has been an increasing volume of biomedical research and translational clinical applications of ^129^XeMRI in various pulmonary pathologies. This is evidence that hyperpolarized gas MRI is emerging at the forefront of clinically relevant lung imaging. The ability of this modality to simultaneously provide data on structure and function is central to its application and provision of research insights into lung pathophysiology. Since 2022, ^129^XeMRI ventilation scans have been approved by the US FDA for clinical use. The appetite for clinical applications of ^129^XeMRI gas exchange has only accelerated since then. In the next few years, the full capacity of ^129^XeMRI (ventilation and gas exchange assessments) is anticipated to be available as a clinical test. Through the exploitation of features detailed in this review and this technology’s non-radioactive and non-invasive advantages, we expect the continued and ongoing application of ^129^XeMRI to research and clinical care.

## Figures and Tables

**Figure 1 diagnostics-15-00474-f001:**
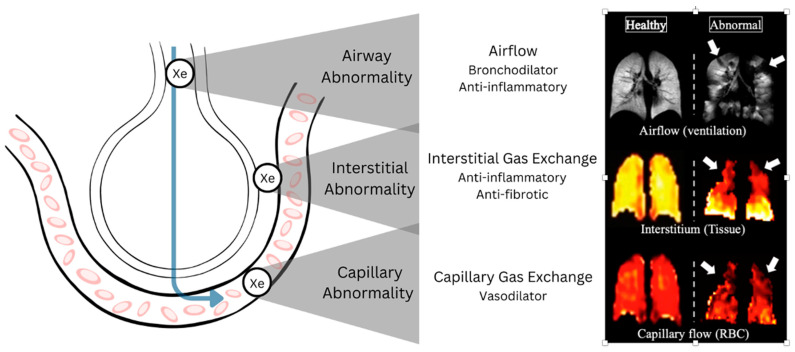
A schematic of the hyperpolarized xenon-129 atom’s movement from airways to alveoli to the interstitium to the red blood cells (RBCs) in the pulmonary capillary vasculature. Three pulmonary anatomic microcompartments are highlighted with gray triangles (airway, interstitium, and pulmonary capillaries), with corresponding classes of therapeutics that can be deployed to address pathologies identified in these microcompartments (bronchodilators, anti-inflammatories, antifibrotics, and pulmonary vasodilators). The left panel illustrates our unpublished examples of healthy (**left**) vs. abnormal (**right**) hyperpolarized xenon-129 MR ventilation, tissue, and RBC images. Uniformly bright lungs in the healthy subject’s ventilation image indicate normal airflow, allowing hyperpolarized xenon-129 gas to reach distal airways. White arrows in the asthmatic subject’s ventilation image indicate no or low ventilation (gray or black) areas. Bright yellow lungs in the healthy subject’s interstitium image indicate normal amounts of hyperpolarized xenon-129 reaching the interstitial tissues, a surrogate parameter of normal interstitial tissue gas exchange. White arrows in the interstitium (tissue) image indicate large dark areas of the right upper and left upper lobes lacking interstitial gas exchange, corresponding to known emphysematous lobes. Bright red lungs in the healthy subject’s capillary flow image indicate normal amounts of hyperpolarized xenon-129 reaching the capillary RBCs, a surrogate marker of normal capillary flow. White arrows in the capillary flow (RBC) images indicate large dark areas of the right upper and left upper lobes lacking capillary blood flow, corresponding to known emphysematous lobes.

**Table 1 diagnostics-15-00474-t001:** Publications using ^129^XeMRI to assess clinical observations or outcomes in pulmonary diseases such as chronic obstructive pulmonary disease (COPD), asthma, cystic fibrosis, idiopathic pulmonary fibrosis (IPF), COVID-19, lymphangioleiomyomatosis (LAM), bronchopulmonary dysplasia (BPD) and other diseases.

Authors	Disease Summary
COPD	
Myc et al. [4]	Significant correlations were found between ^129^XeMRI and FEV1, %DLCO, and emphysema in COPD
Kaushik et al. [5]	^129^XeMRI ADC correlates with pulmonary function, and detects emphysema and age/posture changes
Qing et al. [11]	^129^XeMRI CSSR spectroscopy and DP imaging reliably assess lung function in COPD, revealing thicker septal walls and low RBC-to-TP ratios, indicating poor gas exchange
Qing et al. [12]	^129^XeMRI, with a quick single scan, effectively identifies pulmonary issues in patients with COPD, matching well with CT and gadolinium-enhanced MRI
Ruppert et al. [6]	^129^XeMRI septal wall thickness correlates with DLCO, distinguishing healthy individuals from smokers and individuals with COPD
Doganay et al. [7]	^129^XeMRI gas distribution COV and functional volumes were significantly lower in subjects with COPD vs. healthy subjects.
Rao et al. [8]	Significant differences in ^129^XeMRI VDP, alveolar sleeve depth, total septal wall thickness, ADC, and RBC/TP were found among healthy young, age-matched controls, asymptomatic smokers, and COPD groups
Guan et al. [9]	^129^XeMRI 3D-SBCSI detects ventilation defects, correlates FEV1 with RBC/Gas, and distinguishes between pulmonary diseases
Mummy et al. [10]	^129^XeMRI baseline bar%ref (barrier uptake relative to a healthy reference population) and DLCO correlated with post-treatment changes in ventilation defect; RBC%ref (red blood cell transfer relative to a healthy reference population) decreased in 58.8% of subjects post-treatment
Asthma	
Ebner et al. [13]	^129^XeMRI VDS is significantly higher in airway obstruction, correlates with disease severity, and does not have a location-specific pattern
Ebner et al. [14]	^129^XeMRI imaging detects airway obstructions in asthma, correlates with PFTs, and shows age-related VDP increase
Mussel et al. [15]	VDP and VHI correlated with lung function (FEV1, FEV1/FVC, FEF 25–75%) but not with ACQ7 or eosinophil count; imaging prompted diagnostic re-evaluation in some cases
Qing et al. [11]	There were significant age-related differences in ^129^XeMRI gas transfer: younger asthmatics had lower tissue uptake and higher blood transfer compared to controls; no differences were found in an older group or post-bronchodilator
Svenningsen et al. [16]	Pre-salbutamol, ^129^XeMRI VDP was higher than ^3^HeMRI, with greater post-salbutamol improvement with HP-^129^XeMRI measurement. Both gases showed VDP and ventilation COV reductions post-treatment, with ^129^XeMRI identifying an airway defect not seen with ^3^HeMRI
Kooner et al. [17]	^129^XeMRI helps detect ventilation issues, inflammation, and airflow problems in asthma by visualizing gas exchange and airflow directly in the lungs
Peiffer et al. [18]	Scintigraphy–Xe MRI correlation was higher than SPECT-XeMRI. VDP correlated with FEV1, FEV1/FVC, and FEF 25–75, separating those with asthma and COPD from controls
Safavi et al. [19]	In ^129^XeMRI, asthma had more defects pre-BD, which reduced post-BD, matching healthy participants
Lin et al. [20]	Children with asthma had higher ^129^XeMRI VDP and defects per slice, correlating with increased healthcare use, oral corticosteroids, and reduced lung function (FEV1, FEV1/FVC)
Hall et al. [21]	One guided bronchoscopy (BT) using ^129^XeMRI resulted in a greater reduction in nonventilated lung and fewer asthma exacerbations compared to three unguided BTs, with similar quality of life improvements
Cystic Fibrosis	
Alam et al. [22]	Multiple-breath washout ^129^XeMRI showed high intra-visit and inter-visit repeatability in both healthy subjects and those with CF. CoV fractional ventilation correlated with LCI, highlighting ventilation heterogeneity’s role in early CF
Guan et al. [9]	The 3D-SBCSI detects ventilation defects, correlates FEV1 with RBC/gas, and distinguishes between pulmonary diseases
Marshall et al. [23]	^129^XeMRI and ^3^HeMRI VDP correlated strongly with each other, FEV1, and LCI, showing similar large-scale agreement. However, ^129^XeMRI VDP was more sensitive to subtle ventilation changes in early lung disease than 1H VDP
Kirby et al. [24]	^3^HeMRI ADC detected significant short-term lung changes in CF, correlating with FEV1 and showing more sensitivity than standard tests
Couch et al. [25]	^129^XeMRI VDP measurements showed high agreement between analysts (ICC = 0.99), differentiating healthy and CF groups and correlating with FEV1 and LCI, supporting multi-center trial feasibility
Bannier et al. [26]	^3^HeMRI ventilation defects were present in all patients despite normal spirometry; CPT caused varied defect distribution changes without significantly altering VDI or VF
Couch et al. [25]	^129^XeMRI VDP measurements showed high agreement between analysts (ICC = 0.99), differentiating healthy and CF groups and correlating with FEV1 and LCI, supporting multi-center trial feasibility
IPF	
Wang et al. [27]	^129^XeMRI showed a 188% increase in barrier uptake in IPF, correlating strongly with DLCO and the RBC/barrier ratio (r = 0.94) but not with CT fibrosis scores
Hahn et al. [28]	^129^XeMRI detected improvements in regional gas exchange in patients with IPF treated with antifibrotics after 1 year, while no improvements were seen with conventional therapies
Eaden et al. [29]	^129^XeMRI ADC increased significantly over 12 months in patients with IPF, indicating microstructural disease progression, despite no changes in PFTs. Strong correlations were found between ^129^XeMRI and DLCO/KCO
Qing et al. [12]	^129^XeMRI revealed significant ventilation and gas exchange abnormalities in UIP, including impaired diffusion and an elevated tissue-to-gas ratio, even in patients with normal PFTs
Hahn et al. [28]	^129^XeMRI MRI detected improvements in regional gas exchange in patients with IPF treated with antifibrotics after 1 year, while no improvements were seen with conventional therapies
Chan et al. [30]	SEM accurately estimates acinar dimensions and shows robustness across varying conditions and acinar length scales, validated using He-3 and Xe-129 simulations for healthy and IPF lungs
Stiefer et al. [31]	^129^XeMRI may predict progression in idiopathic pulmonary fibrosis (IPF)
COVID-19	
Grist et al. [32]	^129^XeMRI revealed alveolar diffusion issues in post-COVID-19 patients, despite normal CT scans
Sanders et al. [33]	^129^XeMRI gas transfer remained impaired up to 1 year post-hospitalization in patients with COVID-19, despite normal lung ventilation and no structural abnormalities
Kooner et al. [34]	^129^XeMRI revealed significantly higher VDP in post-COVID, especially in hospitalized participants, correlating with reduced 6MWD and post-exertional SpO2
Matheson et al. [35]	^129^XeMRI effectively detects lung and vascular abnormalities in PACS, aiding COVID-19 diagnosis and management
Eddy et al. [36]	Four distinct long COVID phenotypes were identified using ^129^XeMRI, showing varying patterns of gas exchange and PFTs, highlighting the tool’s ability to differentiate long COVID pathophysiology for personalized care
Kooner et al. [37]	Post-COVID-19 patients showed improved lung function, gas exchange, and quality of life by 15 months. Early ^129^XeMRI VDP predicted exercise gains, and respiratory treatment improved quality of life
LAM	
Walkup et al. [38]	^129^XeMRI detected ventilation deficits in LAM, correlating with FEV1/FVC and DLCO, offering sensitive assessment for screening and management
BPD	
Stewart et al. [39]	^129^XeMRI detected mild ventilation abnormalities and elevated ADC in patients with BPD, demonstrating feasibility for assessing neonatal lung disease
Miscellaneous Disease	
Rankine et al. [40]	^129^XeMRI identified dose-dependent changes in ventilation, membrane uptake, and RBC transfer post-RT, aiding in assessing radiation-induced lung injury

^129^XeMRI—hyperpolarized xenon-129 magnetic resonance image; ^3^HeMRI—hyperpolarized helium-3 magnetic resonance image; CSSR—chemical shift saturation recovery; DP—dissolved phase; COV—coefficient of variation; VDP—ventilation defect percent; VDS—ventilation defect score; ADC—apparent diffusion coefficient; RBC/TP—red blood cells and interstitial tissue/plasma; 3D-SBCSI—3D-single breath chemical shift imaging; BD—bronchodilator; VHI—ventilation heterogeneity index; ACQ7—asthma control questionnaire 7; SPECT-XeMRI—single-photon emission computed tomography xenon magnetic resonance imaging; BT—bronchial thermoplasty; CoVFV—coefficient of variation; CPT—chest physical therapy; VDI—ventilation defects per image; VF—ventilated lung fraction; DW-MRI—diffusion-weighted magnetic resonance imaging; DLCO/KCO—diffusion capacity for carbon monoxide/transfer coefficient of the lung for carbon monoxide; SEM—stretched exponential; 6MWD—six-minute walk distance.

## Data Availability

No raw scientific data are presented in this manuscript.

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
