# Peer review of "Hyperpolarized Xenon-129 MRI: Narrative Review of Clinical Studies, Testing, and Implementation of Advanced Pulmonary In Vivo Imaging and Its Diagnostic Applications"

_diagnostics, 2025, doi:10.3390/diagnostics15040474_

Round 1

Reviewer 1 Report

Comments and Suggestions for Authors

Thanks for the opportunity to review MacLeod and colleagues' manuscript. In it, they outline the clinical advances made by 129XeMRI in recent decades in common pulmonary diseases with high healthcare burdens. 

The manuscript is well written in general and includes primary respiratory disorders where the 129XeMRI can be employed.

I have only minor comments/suggestions for the authors:

Since all the review papers should be comprehensive in their nurture, I recommend avoiding the term in the title and adding "a narrative review" to the end since this alludes to the structure beyond the depth of the content.

The COPD section is really interesting and offers some data about the correlation between PFT variables and VDP. I recommend including numerical variables, which are the values for such correlations.

Finally, consider dividing Table 1 and including the corresponding section of the text.

Reviewer 2 Report

Comments and Suggestions for Authors

This is an interesting narrative review article about clinical utility and diagnostic value of hyperpolarised xenon-129 MRI (XeMRI) in respiratory medicine. The authors describe the basic principles of the test and its application for the structural and functional evaluation of the respiratory system in major pulmonary diseases, including chronic obstructive pulmonary disease, asthma, idiopathic pulmonary fibrosis, and cystic fibrosis, with table 1 comprehensively synopsising the findings from previous studies about the use of XeMRI. The comparison of XeMRI to other imaging modalities of the chest as well as the discussion of the advantages and disadvantages of the method would be useful to be included. Also, the addition of characteristic images of XeMRI in the major pulmonary diseases and in healthy subjects and the report of the main findings would further improve the quality of the article and better familiarise the reader with the imaging method. A well-written and comprehensive article that provides the reader with the current evidence for the use of XeMRI of the chest in clinical practice and research.
